# Crosstalk between Oxidative Stress and Inflammation Caused by Noise and Air Pollution—Implications for Neurodegenerative Diseases

**DOI:** 10.3390/antiox13030266

**Published:** 2024-02-22

**Authors:** Marin Kuntić, Omar Hahad, Thomas Münzel, Andreas Daiber

**Affiliations:** 1Department of Cardiology 1, University Medical Center of the Johannes Gutenberg University, 55131 Mainz, Germanyomar.hahad@unimedizin-mainz.de (O.H.); 2German Center for Cardiovascular Research (DZHK), Partner Site Rhine-Main, 55131 Mainz, Germany

**Keywords:** environmental risk factors, air pollution, transportation noise exposure, oxidative stress, inflammation, neurodegeneration

## Abstract

Neurodegenerative diseases are often referred to as diseases of old age, and with the aging population, they are gaining scientific and medical interest. Environmental stressors, most notably traffic noise and air pollution, have recently come to the forefront, and have emerged as disease risk factors. The evidence for a connection between environmental risk factors and neurodegenerative disease is growing. In this review, the most common neurodegenerative diseases and their epidemiological association with traffic noise and air pollution are presented. Also, the most important mechanisms involved in neurodegenerative disease development, oxidative stress, and neuroinflammation are highlighted. An overview of the in vivo findings will provide a mechanistic link between noise, air pollution, and neurodegenerative pathology. Finally, the importance of the direct and indirect pathways, by which noise and air pollution cause cerebral damage, is discussed. More high-quality data are still needed from both epidemiological and basic science studies in order to better understand the causal connection between neurodegenerative diseases and environmental risk factors.

## 1. Introduction

### 1.1. Air and Noise Pollution—Overview

As the world is becoming more and more industrialized, and medicine and sanitation are becoming better in combating communicable diseases, the scientific community is focusing on risk factors of the physical environment, such as air and noise pollution [1,2]. Air pollution is a broad term used to describe the toxic gases, liquids, and solids derived from natural and artificial sources present in the atmosphere [3]. According to the Global Burden of Disease study, air pollution ranked 4th among the leading causes of global morbidity and mortality for both males and females [4]. The global mortality burden from air pollution was assessed by multiple models and recently reached 8.7 million to 10.2 million annual deaths, accounting for almost 20% of all deaths worldwide [5,6]. Solid constituents of air pollution, commonly known as particulate matter (PM), recently gained significant interest because of its prevalence in developing and industrialized countries, complementing gaseous air pollution, like ozone and nitric oxides, which have been studied for many decades [3,7]. Apart from the composition of PM, its size is a major determinant of its toxicity for human organ systems [8,9].

Traffic noise has become a prominent modern risk factor for human disease, making the famous words of the Nobel Prize laureate Robert Koch prophetic: “One day man will have to fight noise as relentlessly as plague and cholera” [10]. The World Health Organization (WHO) estimates that at least 1.6 million healthy life years are lost yearly due to traffic-related noise in Western Europe alone, and that 40% of people in the European Union live in areas where traffic-related noise exceeds the recommended thresholds [11]. Although traffic noise is recognized as a prominent risk factor, it was not included in the latest report of the Global Burden of Disease study (only occupational noise was mentioned there, rather at the bottom of the ranking [12]).

In 2005, Christopher P. Wild coined the term “exposome” in order to describe all of the environmental exposures that influence human physiology over the duration of a lifespan [13]. This term does not only include external stressors such as climate, pathogens, UV radiation, noise, and air pollution but also sociological factors such as lifestyle and socioeconomic status, psychological factors such as anxiety and depression, and daily surroundings, such as urban environment and green spaces [14,15,16]. Exposome research is important, as it provides an important link between disease onset and progression and human endeavors. It provides a rational basis for the statement “Genetics load the gun, but environment pulls the trigger” [17,18] and the estimation that 2/3 of all chronic diseases have environmental instead of genetic causes [19].

### 1.2. Neurodegenerative Disease—Overview

Advances in medicine and hygiene during the past century have significantly contributed to eradicating communicable diseases. Today, due to the aging population, non-communicable diseases such as cardiovascular diseases (CVD) and neurodegenerative diseases, together with cancer, represent the major contributors to the global disease burden [20]. Neurodegenerative diseases are usually referred to as diseases of old age, as they are most prevalent in the elderly population [21,22]. It is estimated that 50 million people globally suffer from dementia and that this number will increase to 130 million by 2050 [23,24]. The most common neurodegenerative diseases are Alzheimer’s disease (AD), Parkinson’s disease (PD), and amyotrophic lateral sclerosis (ALS), and they will be the focus of this review. Other neurodegenerative diseases, like multiple sclerosis, Huntington’s disease, multiple system atrophy, and frontotemporal dementia, are also relatively common, and will be addressed where appropriate.

AD is the most common type of dementia and the most common neurodegenerative disease, with a prevalence of almost two-thirds of all dementia cases [25]. As some reports predict that dementia will almost double by the year 2050 [26], strategies to prevent and alleviate AD will become essential for reducing the disease burden. AD is characterized by loss of memory and general cognitive decline, resulting in aggression, confusion, problems with attention and language, mood-swings, and sleep impairment [27,28,29]. The prognosis for AD patients is poor with only 5–9 years of life left after diagnosis [30]. A diagnosis of AD severity usually follows a classic dementia seven-stage progression model, starting with no impairment and moving through stages of cognitive decline, all the way to late-stage dementia [31]. The disease is often identified late, when patients notice an ongoing cognitive impairment. The presence of AD as a subtype of dementia was previously only possible by post-mortem autopsy, as clear atrophy of the frontal and temporal cortices is visible [32]. Modern diagnostic tools, including magnetic resonance imaging (MRI), and blood and spinal fluid biomarkers, improved early diagnosis of AD [33,34]. There are several mechanistic concepts of AD pathophysiology, and the most widely accepted one is the aggregation of the extracellular amyloid beta (Aβ) protein plaques and the formation of the intracellular neurofibrillary tangles (NFTs) of tau protein [35]. These protein plaques inhibit synapse communication, promote inflammation through microglia activation, and impair blood flow in the brain (cerebral amyloid angiopathy) [36].

PD is a syndrome with different underlying causes and variable symptoms. Still, it is overall characterized by the loss of the motor neuron function, resulting in tremors, rigidity, slowness and difficulty in movement, and impairment in posture [37]. There are currently over 10 million people living with PD worldwide, and the number of newly diagnosed patients is rising every year [38]. Unlike other neurodegenerative diseases, PD patients can live with the symptoms for a long time, as the prodromal period is substantially longer than in other types of neurodegenerative diseases [39]. Only a small percentage (5–10%) of all PD cases are inherited, although results from the genome-wide association study have identified 90 independent genome-wide significant risk signals [40]. Most of the PD cases occur sporadically (90–95%). Symptoms of PD are a result of the death of neurons in the substantia nigra, a region of the midbrain that supplies dopamine to the basal ganglia [41]. The pathology of PD is defined by Lewy bodies, an accumulation of α-synuclein protein inside neurons [42]. Misfolded α-synuclein was also found in AD patients, but it is still not recognized as a major cause of AD pathology, as it is for the PD pathology [43]. Lewy bodies consist not only of α-synuclein aggregates but also contain lipids in the form of membrane fragments, mitochondria, and vesicular structures [44]. PD is usually treated with dopamine replacement, which suppresses the symptoms, but no cure is available [45].

ALS, also known as Lou Gehrig’s disease, results in the progressive loss of motor neurons in the cerebral cortex, brainstem, and spinal cord [46]. ALS patients suffer from muscle wasting, weakness, and paralysis, leading to death by respiratory failure [47]. ALS is relatively rare in comparison with AD and PD, with incidence rate of 10–15 cases per 100,000 capita [48]. The progression of ALS is rapid, with death occurring 2–3 years after diagnosis [49]. Only 10% of all cases are familial, but mutations in up to 40 gene mutations have been associated with ALS [50]. Due to such a wide variety of different possible genetic causes, there is no known cure for ALS, only therapies that alleviate symptoms, such as antioxidant therapies and treatments for dysphagia and respiratory failure [51]. More than 20 years ago, mutation of the superoxide dismutase (*SOD1*) gene was identified as the major cause of the familial ALS [52], but other genes, such as fused in sarcoma (*FUS*), transacting response DNA-binding protein (*TARDBP*), and chromosome 9 open reading frame 72 (*C9ORF72*), were also shown to be accountable for the large percentage of familial ALS cases [53]. The pathology of ALS also includes aggregations of ubiquitinated proteins in the cytoplasm of degenerating motor neurons and surrounding oligodendrocytes [54].

## 2. Role of Oxidative Stress and Inflammation in Neurodegenerative Diseases

Neurodegenerative diseases are most often characterized by the accumulation of protein aggregates in either the intracellular or the extracellular space of the neuronal system [55]. This accumulation of protein aggregates is accompanied by neuroinflammation [56,57,58] and oxidative stress [27,59,60]. Oxidative stress is a pathophysiological state in which the amount of reactive oxygen and nitrogen species (RONS) produced exceeds the capacity of the system to remove them successfully or to repair the resulting oxidative damage [61,62]. This imbalance happens due to many factors but is ubiquitous in almost all human diseases [63,64]. Inflammation of the nervous tissue is a hallmark of neurodegenerative diseases [56]. Inflammation and oxidative stress are almost inseparable and exhibit a crosstalk [65] that has been explored deeply for many years in both human and animal models [57,66]. Immune cells that infiltrate tissue produce large amounts of RONS as a host defense mechanism, and oxidative stress in tissues is a signal for the recruitment of pro-inflammatory immune cells [67]. Therefore, exploring oxidative stress and inflammation in neurodegenerative pathophysiology as an intertwined pathomechanism is important. The general interplay between neuroinflammation and oxidative stress is shown in Figure 1. It is important to keep in mind that neuroinflammation is not the only source of cerebral oxidative stress, and that other prominent sources like the mitochondria and the dopamine metabolism can significantly contribute to neurodegeneration [68,69]. The RONS that we talk about in the subsequent sections, are mostly those produced from NADPH oxidases during inflammation and by mitochondria upon stress challenges (superoxide and hydrogen peroxide) or direct physicochemical effects from air pollutants (e.g., nano-sized combustion-derived particles). The term RONS also comprises the major product of superoxide and nitric oxide, peroxynitrite. Besides these most important RONS, there may be some minor contribution by hydroxyl radicals that are directly formed from transition metal reactions with peroxides or other redox-active chemicals on the surface of air pollution particles. More details on the species formed and their sources can be found in recent overviews [3,70].

### 2.1. Neurodegenerative Disease and Oxidative Stress

Although the brain constitutes only around 2% of the total body mass, it is responsible for consuming 20% of the available oxygen [71]. The high oxygen consumption is needed to produce ATP in order to maintain the neuronal membrane potential through Na^+^ and K^+^ gradients [72]. Mitochondrial respiratory chain, responsible for most oxygen consumption, can leak electrons to reduce molecular oxygen and form superoxide anion radical (^•^O_2_^−^) [73]. Although there are mechanisms to eliminate the ^•^O_2_^−^ from the cell via enzymatic catalysis by SOD enzymes, the kinetically favorable reaction of ^•^O_2_^−^ with nitric oxide (^•^NO) will ultimately lead to peroxynitrite (ONOO^−^) formation before ^•^O_2_^−^ can be eliminated [74]. ^•^NO is produced by nitric oxide synthases (NOS) that are expressed as different isoforms, e.g., by endothelial cells (eNOS) and neuronal cells (nNOS), or can be induced (iNOS) in immune cells, such as macrophages and microglia, but also other cell types, such as astrocytes [75,76]. Free radicals are important at physiological concentrations as they can act as cellular signaling molecules. Still, during oxidative stress, they are produced at high levels and can damage lipids, proteins, and nucleic acids. As the cerebral tissue is rich in functional lipids it is especially susceptible to oxidative stress damage [77].

During many years of AD research, oxidative stress markers have often been associated with AD pathology, especially by using post-mortem brain material of AD patients. In these brain samples, researchers identified elevated activity of glucose-6-phosphate dehydrogenase and 6-phosphogluconate dehydrogenase, pointing to the increased demand for reducing equivalents in the form of NADPH for brain peroxide metabolism [78]. Classical markers of lipid peroxidation, such as malondialdehyde (MDA), 4-hydroxynonenal (4-HNE), and F2-isoprostanes were also observed in AD patient brains [79,80]. It was even proposed that these lipid peroxides are responsible for the observed AD protein aggregates by creating protein adducts [81]. Evidence of protein and nucleic acid damage from oxidative stress in AD patients was envisaged by protein carbonylation and an increase in 8-hydroxyguanosine (8-OHG) levels in brain homogenates [82,83]. Increased 3-nitrotyrosine (3-NT) levels, a marker of ONOO^−^-mediated damage, were observed in AD patient brains and cerebral blood vessels, leading to the hypothesis that Aβ plaques can alter iNOS and eNOS expression [84,85,86]. The AD-associated oxidative stress is also accompanied by lower reduced glutathione levels (mostly measured by a diminished GSH/GSSG ratio) and reduced expression of antioxidant enzymes, such as SOD enzymes, catalase (CAT), and glutathione peroxidases (GPx) [87,88,89]. Most of the antioxidant therapies used to combat AD progression failed, and many of the drugs used to reduce cognitive decline did not provide solid evidence of oxidative stress reduction, pointing to the possibility that the oxidative stress is a symptom of AD pathology rather than a cause [90].

Similarly to AD, markers of oxidative stress were also identified in PD patients, but they were mostly localized in the substantia nigra [41]. Increased levels of MDA, 4-HNE, and acrolein-modified proteins were all observed in the substantia nigra, frontal cortex, and brainstem but not in other brain regions, such as the caudate nucleus and putamen [91,92,93,94,95]. This is important as 4-HNE adducts with α-synuclein protein were shown to promote aggregation, a hallmark of PD [96]. Also, nitration of α-synuclein was suggested as an oxidative mechanism promoting aggregate formation [97,98]. Other markers of oxidative stress, such as protein carbonyls, 8-OHG and 8-OHdG were also identified in substantia nigra, pointing to direct protein and nucleic acid oxidation [99,100]. Interestingly, some research shows that protein carbonyls might come from L-DOPA, a therapeutic drug used to treat dopamine deficiency in PD, as it can have pro-oxidant properties via hydroxyl radicals [101]. L-DOPA could also provide the substrates for mitochondrial monoamine oxidases, e.g., dopamine and other catecholamines, with subsequent hydrogen peroxide formation, which was therapeutically exploited using monoamine oxidase inhibitors in PD patients [102]. Another prominent feature of dopamine is its ability to undergo auto-oxidation, or metal-oxidation reactions, resulting in formation of dopamine quinones [103]. The short-lived dopamine-o-quinone is created together with ^•^O_2_^−^, and can undergo further oxidation to form aminochrome [104]. Both dopamine-o-quinone and aminochrome are highly reactive and form protein adducts on the cysteine, lysine, and tyrosine residues, causing specific damage to mitochondria and thereby worsening the PD pathology. An increase in oxidative stress in PD patients is also accompanied by a decrease in GSH and an increase in SOD levels, specifically in substantia nigra [41,105,106]. The involvement of mitochondria was also observed in PD patients with a reduction in respiratory chain complexes activity [107,108], and inhibition of complex I, e.g., by rotenone, was previously shown to cause Parkinsonism [109], which was even used as an animal model of PD. Therapies with coenzyme Q10, an important molecule for the function of the mitochondrial respiratory chain, showed limited but positive outcomes on the progression of PD [110], although later meta-analysis showed no improvement compared to placebo [111].

ALS is tightly related to oxidative stress as one of the major mutations associated with familial ALS is on the *SOD1* gene [53]. A connection between the impaired function of SOD1, peroxynitrite formation, and ALS was recognized early [52]. The mutated gene can produce misfolded proteins that create aggregates and lose their catalytic function, rendering the enzyme unable to clear ^•^O_2_^−^, thus producing oxidative stress [112,113]. Other mutated genes identified in ALS patients can also contribute to oxidative stress. The protein encoded by the *TARDBP* gene is known to affect the nuclear factor erythroid 2-related factor 2 (Nrf2) pathway, an important antioxidant response mechanism [114,115]. The protein encoded by the *FUS* gene is involved in the repair of DNA-double strand breaks, and *C9ORF72* gene mutation can produce dipeptide repeat proteins that are responsible for the interruption of mitochondrial membrane potential and loss of DNA repair [116,117]. Like in AD and PD patients, markers of oxidative damage of proteins, lipids, and nucleic acids were found in ALS patients. 4-HNE and 3-NT were observed to be increased in the cerebrospinal fluid of ALS patients together with 8-OHdG [118,119,120,121]. Although the levels of oxidative stress markers were not associated with the severity of the ALS progression, some studies have directly shown increases in oxidative stress in certain brain regions of ALS patients [122,123]. Similarly to the AD and PD, almost all antioxidant therapies have failed to produce a meaningful improvement of symptoms or prolonged life [124]. Interestingly, the ALS drug edaravone, which was proposed to confer its pharmacological action through free radical and peroxynitrite scavenging, was shown to improve patient conditions, but only in a group with a high vital capacity and early-stage disease diagnosis [125].

### 2.2. Neurodegenerative Diseases and Inflammation

Inflammation is a hallmark of all neurodegenerative diseases as the accumulation of misfolded proteins into plaques can activate pathogen-associated and damage-associated molecular patterns (PAMPs and DAMPs) leading to sterile inflammation [126,127]. Some of the most prominent receptors tasked with these pattern recognitions in the nervous system are the Toll-like receptors (TLRs), which are present in microglia and astrocytes [128]. Activated TLRs mediate the release of both pro- and anti-inflammatory cytokines, which can either help with the repair of tissue damage and resolution of inflammation in the acute setting or promote further damage and oxidative stress in a chronic inflammatory setting [129,130]. The TLR signaling activates transcription factors, like activator protein 1 (AP-1) and nuclear factor ‘kappa-light-chain-enhancer’ of activated B-cells (NF-κB), that transcribe genes for pro-inflammatory cytokines [131]. Another class of danger sensing receptors, the pattern recognition receptors (PRRs), are located in the cytoplasm where they can be activated by the misfolded protein accumulation. These cytoplasmic PRRs are often called the inflammasome (most notably the nucleotide-binding domain and leucine-rich repeat-containing receptors family pyrin domain containing 3 (NLRP3)). They can activate pro-inflammatory pathways, e.g., via IL-1β [56,132]. Since the presence of misfolded protein aggregates produces a response where the microglia try to clear the cell debris through M2 polarization and release of anti-inflammatory cytokines, the long-lasting presence of protein plaques and the failure of microglia to effectively clear them lead to pro-inflammatory M1 polarization [133]. These long-lasting protein plaques are a hallmark of AD and PD, providing ideal conditions for chronic inflammation development. Activated microglia express iNOS that can, together with the activation of the (phagocytic) NADPH oxidase, produce peroxynitrite anion that nitrates proteins [134]. This was indeed observed in the brains of AD patients, pointing to a strong immune cell activation [135,136]. It was previously observed that Aβ could initiate the NF-κB sterile inflammation pathway though through the activation of a cluster of differentiation 36 (CD36), further promoting TLR4 and TLR6 activity [137]. CD36 can also promote Aβ aggregation and induce NLRP3 inflammasome, increasing IL-1β release [138]. IL-1β can later increase the activity of p38 mitogen-activated protein kinases (p38-MAPK) and glycogen synthase kinase 3 (GSK-3), promoting the phosphorylation of the tau protein, another hallmark of AD [139]. In PD, the presence of α-synuclein can also directly stimulate the NLRP3 inflammasome [140,141]. Importantly, ROS derived from mitochondria [142] or from NADPH oxidase [65] can activate the NLRP3 inflammasome by redox-dependent pathways, including thioredoxin-interacting protein (TXNIP) as a central redox switch [143].

The incidence of multiple sclerosis is similar to that of ALS, but multiple sclerosis is considered the most common type of chronic inflammatory disease of the CNS [56]. Multiple sclerosis is characterized by demyelination of neurons, inhibiting signal transduction [144]. Damaged myelin sheets are taken up and degraded by activated resident macrophages, creating a multiple sclerosis lesion [145]. These lesions typically consist of CD3^+^/CD8^+^ T lymphocytes and of CD68-positive microglia, and oxidized lipids [146,147]. In multiple sclerosis, macrophages and other circulating immune cells can cross the blood–brain barrier, creating an adaptive immune response, further damaging the neuronal tissue and promoting inflammation [148].

ALS is a systemic disease that attacks the motor neurons. Thus, it is outside of the brain tissue itself, making the adaptive immune response a bigger contributor to the overall inflammation burden [149]. Many circulating markers of inflammation were observed in ALS patients. Still, individually, these markers often fail to provide a good measurement for the severity and the progression of the disease [150]. Most of the many mutated genes in ALS are not directly related to the immune system but can indirectly increase inflammation, mostly through the NF-κB pathway. Transactive response DNA-binding protein of 43 kDa (TDP-43) is observed to form plaques in sporadic ALS patients after phosphorylation in the cytoplasm [151]. The TDP-43 is a known activator of p65, a subunit of NF-κB that promotes pro-inflammatory cytokine release [152]. Another activator of NF-κB is the FUS protein, known to be mutated in familial ALS [153]. The FUS protein, together with SOD1 and C9orf72, which carry known ALS-associated mutations, are more detrimental, and induce a stronger inflammatory response when expressed in glia cells than in neuronal cells, pointing to an important role of resident macrophages in ALS neurodegeneration [154,155,156,157].

It is important to remember that oxidative stress can induce (sterile) inflammation or exacerbate inflammatory pathways at various levels (see Figure 2 for an overview). Hydrogen peroxide and peroxynitrite cause redox modifications in central pro-inflammatory molecules such as high mobility group box 1 (HMGB1) protein, inflammasome constituent NLRP3, *NF-κB*, and the process of NETosis (reviewed in [65]). Also, many of the up-stream regulators of these master switches of inflammation are redox-regulated as exemplified by reversible and irreversible sulfoxidation, S-nitros(yl)ation, S-glutathionylation, and protein tyrosine nitration of p38, p50, p65, S100A9, thioredoxin (Trx), IkB, and TXNIP (reviewed in [65,158]). Of note, NOX-2 activation, per se, is indispensable for the activation, recruitment, and infiltration of myelomonocytic cells [159,160] and T-cells [161]. In addition, the migration of monocytes/recruitment of macrophages is controlled by the cellular redox potential [162], and potential redox regulatory mechanisms involve redox modifications of MAPK phosphatase 1 [163] and Slingshot-1L-binding protein 14-3-3zeta [164] as well as changes in the S-glutathionylation pattern [165]. By this adverse redox signaling of ROS formed by damaged mitochondria or by activated phagocytic cells, inflammatory reactions may be initiated and aggravated. Conversely, inflammation causes cell activation, which is always associated with ROS formation by cellular ROS sources. Specifically, activated phagocytic cells infiltrating into the tissue can induce an oxidative burst by substantial superoxide formation from phagocytic NADPH oxidase. This crosstalk between oxidative stress and inflammation may create a vicious circle [65,158], leading to substantial tissue damage and neurodegeneration when happening in the brain.

Since microglia play a central role in neuroinflammation, it is important to consider them in the light of novel findings associating their function with neurodegenerative diseases [166]. Advances in single cell RNA sequencing revealed that the traditional M1/M2 polarization paradigm is not detailed enough to explain the intricate differences in specific microglia gene expressions associated with neurodegenerative diseases [167]. Microglia responding to Aβ have a distinct expression profile envisaged by upregulation of *APOE*, *TREM2*, *GPNMB*, and are CD163-positive [168,169,170]. A similar specific gene expression was found for phosphorylated tau protein, like *GRID2* [168], and for α-synuclein, like *IL1B*, *GPNMB*, and *HSP90AA1*, in PD patients [171].

Astrocytes are part of the glymphatic system that is responsible for clearing protein plaques, such as Aβ [172], but they are also involved in neurodegenerative disease-induced neuroinflammation [173]. Similarly to microglia, astrocytes express neurodegenerative disease-associated genes that tend to promote inflammatory processes, like *CD44*, *GFAP*, and *HSPB1*, observed in PD, Huntington’s disease, and AD [171,174,175,176,177].

## 3. Effects of Noise and Air Pollution on Inflammation and Oxidative Stress in Neurodegenerative Disease—Mechanistic Insights from Animal Studies

The intricate link between oxidative stress and neuroinflammation can mostly be explained through microglia activation. Microglia as resident macrophages of cerebral tissues can be both, stimulated to produce ROS, and sense ROS as secondary messengers to promote pro-inflammatory behavior [178]. Many in vitro and in vivo studies have shown that air and noise pollution can affect neuroinflammation and oxidative stress, thereby potentially influencing the onset and progression of neurodegenerative diseases. In this chapter, we will focus on the preclinical research showing mechanistic links between neuroinflammation and oxidative stress in response to air and noise pollution.

### 3.1. Air Pollution

Air pollution is now an established risk factor in many diseases, including neurodegenerative diseases [4]. Many of the components of air pollution were shown to be associated with neuroinflammation and oxidative stress. Particulate matter (PM), the solid component of air pollution, induced neuroinflammation in mouse models [179,180,181]. In these studies, neuroinflammation was mostly observed through the increase in pro-inflammatory cytokines in the brain tissue, such as TNF-α, IL-1α/β, and IL-6, and the activation of NF-κB. Exposure of mice to ultrafine PM, with a diameter of less than 0.1 µm, also showed an increase in neuroinflammation through the upregulation of TNF-α, iNOS [182], cytokines/chemokines IL-1β, IL-6, TGF-β, CCL5, and CXCL1 in the hippocampus [183]. This upregulation of pro-inflammatory cytokines was accompanied by increased oxidative stress, envisaged by higher levels of 4-HNE and 3-NT-positive proteins. Exposure to PM was also associated with known pathomechanisms of neurodegenerative diseases, also including mitochondrial dysfunction and reactive oxygen species formation [7]. Studies in mice and rats showed that ambient PM exposure can increase tau phosphorylation [184,185], elevate beta-amyloid precursor protein (β-APP) and Aβ(42) oligomers [186,187], and increase plaque deposition [188,189]. PM exposure also resulted in cognitive decline, measured as learning impairment and a decrease in memory retention [190,191,192]. Ozone, another prominent gaseous component of air pollution, was observed to induce neuroinflammation and oxidative stress. A study in rats observed increased NF-κB and cytochrome c and COX-2 levels in substantia nigra, after ozone exposure, indicating potential implications for PD [193]. Another study from the same group also found a p53-dependent stress response and a decrease in dopaminergic neurons in the substantia nigra, together with an increase in circulating markers of lipid peroxidation [194]. Implications for AD were also found in an ex vivo study on rats where microglia showed proinflammatory activation, and Aβ(42) aggregation was increased in response to ozone exposure [195]. Aβ(42) accumulation was also observed in the mitochondria of the rat hippocampus after ozone exposure for 60 and 90 days [196].

In addition to exposure to individual air pollution components, some studies examined the influence of concentrated air or naturally occurring air pollution. Pioneering studies on dogs in Mexico showed that high amounts of air pollution increase levels of NF-κB and iNOS in multiple brain regions, together with disruption in blood–brain barrier, degradation of cortical neurons, and accumulation of neurofibrillary tangles, and Aβ(42) in the olfactory bulb and frontal cortex [197,198]. The authors also demonstrated that administration of the anti-inflammatory drug nimesulide improved 3-NT levels, pointing to reduced contribution from iNOS, whereas no changes in other parameters, including AD pathology, were observed [199]. In addition to concentrated ambient air pollution, exposure studies with diesel exhaust PM and full vehicle emissions showed similar microglia activation and an increase in neuroinflammation and oxidative stress in multiple brain regions [200,201,202]. Interestingly, there are some indications that alternative fuel emissions, like those from biodiesel, do not initiate neuroinflammation but rather activate microglia in a different manner, causing no neuronal damage [203], although the evidence provided by these studies is still preliminary. In animal models of neurodegenerative diseases, like AD [184,189,190,204] and PD [205,206], air pollution was shown to aggravate the progression and severity of the disease.

Air pollution has two potential pathways by which it can reach the neuronal system: a direct pathway where components of air pollution can reach brain tissue directly, and an indirect pathway where the influence of air pollution components (or associated toxins) in other organs (mostly lung as a first point of entry) gains a systemic character that finally reaches brain tissue [207]. In the direct pathway, air pollution-derived PM can enter the brain through the olfactory bulb [208], or ultrafine (nano-sized) PM can cross the air–blood barrier in the lung and enter circulation, later reaching the brain [209,210]. In our research, we have indirectly observed the translocation of ambient air pollution-derived PM into the circulation, as the exposure of mice resulted in an impairment of microvascular function and an increase in oxidative stress in cerebral and retinal arterioles [211]. In the indirect pathway, air pollution components cause oxidative stress and inflammation in the lung, which then spreads secondary messengers through pro-inflammatory cytokines and pro-oxidants to the systemic circulation, interrupting the blood–brain barrier integrity and inducing microglia activation in the brain [212,213,214]. Our observations align with these pathways as well, as many circulating cytokines were elevated in mice after PM exposure [211]. In addition to circulating cytokines, a non-cytokine-dependent pathway can result in microglia activation and neuroinflammation, most notably through the neurohormonal stress response by activation of the sympathetic nervous system and hypothalamus–pituitary–adrenal (HPA) axis [195,215]. An overview of the mechanisms by which air pollution affects neuroinflammation and oxidative stress is shown in Figure 3.

### 3.2. Transportation Noise

Traffic noise has only recently received the mechanistic background to better understand the epidemiological data on associations of noise with disease [70]. Like air pollution, noise also has two pathways by which it affects human health: the direct pathway, where loud noise causes inner ear damage and hearing loss as well as sleep deprivation, and the indirect pathway, where noise at lower sound pressure levels impairs daily activities, induces annoyance and disturbs sleep, causing elevated stress [216]. It is through the indirect pathway that most people are exposed to noise. This noise annoyance can trigger HPA axis signaling and sympathetic nervous system activation, leading to the release of glucocorticoids and catecholamines, dysregulation of the circadian rhythm, and increased inflammation and oxidative stress [74,217,218,219]. All of this can then initiate cerebral inflammation and promote the progression of neurodegenerative diseases [220]. An overview of the mechanisms by which noise affects neuroinflammation and oxidative stress is shown in Figure 4.

Fewer data exist on the effects of noise on neurodegenerative diseases than on the effects of air pollution. Nevertheless, some animal studies have shown that noise exposure is related to neuroinflammation, oxidative stress, and neurodegenerative disease pathology [221]. Neuronal inflammation was also shown in our studies in mice, where iNOS, CD68, and IL-6 were elevated after 4 days of noise exposure (mean sound pressure level of 72 dBA) [222]. This neuronal inflammation was accompanied by elevation in oxidative stress derived from phagocytic NADPH oxidase (NOX2), pointing to an immune system activation. Evidence for a systemic inflammation was observed as well [223]. Another well-established effect of noise exposure is the increase in tau phosphorylation, as observed in multiple studies [224,225,226]. Noise exposure was also shown to disturb the AMPK-mTOR pathway, impeding the autophagosome–lysosome fusion and resulting in autophagosome aggregation [226], another hallmark of AD pathology [227].

Studies in animal models of neurodegenerative disease also show a detrimental impact on disease progression. In a senescence-accelerated mouse prone 8 (SAMP8) model, the exposure to high sound pressure noise caused increased Aβ(42) accumulation and higher levels of tau phosphorylation [228]. This worsening of AD pathology was accompanied by upregulation of AD-associated genes such as Arc, Egr1, Egr2, Fos, Nauk1, and Per2. The impact of noise exposure on AD pathology was confirmed by other studies using the SAMP8 model [229,230]. A study conducted in the APP/PS1 Tg mouse model of AD showed reduced hippocampal tight junction protein levels. It increased circulating pro-inflammatory markers in the noise-exposed group [231]. In a study on gestational noise exposure, APP^NL-G-F/NL-G-F^ mice showed an activation of the HPA axis, impairment in spatial learning, and Aβ(42) deposition [232]. Interestingly, the effects were more pronounced in female than male mice. In addition to the indirect pathway of noise-induced detrimental effects, a study applying noise with high sound pressure levels found associations of noise with an increase in c-Fos, c-Myc, and β-APP, pointing to the importance of inner ear damage and hearing loss in the progression of AD [233]. The importance of hearing loss for the propagation of AD-like pathophysiology was also confirmed by other noise exposure studies with high sound pressure levels [234,235,236].

## 4. Environmental Stressors and Neurodegenerative Disease—Evidence from Epidemiological Studies

Environmental stressors, such as transportation noise and air pollution, have previously been linked to neurodegenerative diseases [207,237,238]. Although the molecular mechanisms are not fully understood, the epidemiological data from both large and small cohorts provide insights into the association between environmental stressors and neurodegenerative diseases. Animal data on the relationship between the onset of neurodegenerative diseases and noise/PM exposure are mostly correlative in nature and large clinical cohorts or epidemiological studies are scarce. Therefore, the present topic is still under debate, and high-quality human research is urgently needed.

### 4.1. Air Pollution and Neurodegenerative Diseases

The available evidence regarding the potential association between air pollution and PD is constrained, and the outcomes lack a consistent consensus. A recent comprehensive meta-analysis, encompassing a total of 21 studies, investigated the connection between long-term exposure to air pollution, second-hand smoke, and the onset of PD. The analysis yielded predominantly non-significant findings with only marginal elevations in risk per 10 μg/m^3^ increments in PM_2.5_ (relative risk [RR] 1.08, 95% confidence interval [CI] 0.98–1.19), ^•^NO_2_ (RR 1.03, 95% CI 0.99–1.07), O_3_ (RR 1.01, 95% CI 1.00–1.02), and CO (RR 1.32, 95% CI 0.82–2.11) [239]. Strikingly, exposure to second-hand smoke was paradoxically associated with a significantly decreased risk of developing PD.

Consequently, another meta-analysis, encompassing 15 studies, revealed slightly elevated risks for the incidence of PD following long-term air pollution exposure, with RRs of 1.06 (95% CI 0.99–1.14) for PM_2.5_, 1.01 (95% CI 0.98–1.03) for ^•^NO_2_, 1.01 (95% CI 1.00–1.02) for O_3_, and 1.34 (95% CI 0.85–2.10) for CO. Furthermore, there was a RR 1.03 (95% CI 1.01–1.05) for hospital admission due to PD in response to short-term exposure to PM_2.5_, although there was considerable heterogeneity among the studies [240].

A more robust effect estimate for the risk of PD emerged from a meta-analysis involving 10 studies, with RRs of 1.06 (95% CI 1.04–1.09) for NOx, 1.65 (95% CI 1.10–2.48) for CO, 1.01 (95% CI 1.00–1.03) for ^•^NO_2_, and 1.01 (95% CI 1.00–1.02) for O_3_. However, it is important to note that this analysis was associated with a high risk of bias [241].

The potential role of PM_2.5_ as a determining factor in the development of dementia has been examined in two recent meta-analyses. In the first analysis, which incorporated data from four cohort studies conducted in Canada, Taiwan, the UK, and the US and included over 12 million elderly individuals aged 50 years or older, a noteworthy threefold increase in the risk of dementia was observed (hazard ratio [HR] 3.26, 95% CI 1.20–5.31) for each 10 μg/m^3^ increase in long-term PM_2.5_ exposure [242]. Subgroup analyses within this study further unveiled an almost fivefold elevated risk for AD (HR 4.82, 95% CI 2.28–7.36).

A comprehensive meta-analysis, encompassing 80 studies from 26 different countries, aimed to investigate the impact of PM_2.5_ on various cerebrovascular and neurological disorders [243]. The findings of this analysis revealed that long-term exposure to PM_2.5_ was associated with an increased overall risk of dementia (odds ratio [OR] 1.16, 95% CI 1.07–1.26), with a particular emphasis on AD (OR 3.26, 95% CI 0.84–12.74). Additionally, increased risks were observed for autism spectrum disorder (OR 1.68, 95% CI 1.20–2.34), PD (OR 1.34, 95% CI 1.04–1.73), and stroke.

Interestingly, recent investigations have highlighted the significant role of cardiovascular diseases in modifying and mediating the association between air pollutants and dementia risk [244,245]. In a prospective study from Sweden, it was observed that the presence or development of heart failure, ischemic heart disease, and stroke (which collectively explained 49.4% of air pollution-related dementia cases) appeared to amplify the link between long-term exposure to PM_2.5_ and NOx and the risk of dementia. This phenomenon is likely due to shared pathophysiological pathways through which air pollutants exert adverse effects on cardiovascular and neurological systems. The study’s authors concluded that as cardiovascular diseases accelerate cognitive decline and anticipate the onset of dementia, exposure to air pollution may indirectly exert detrimental effects on cognition, by pathways associated with cardiovascular disease, without the pollutants necessarily reaching the brain.

Substantiating this perspective, a prospective study from Italy demonstrated that the neurological effects of air pollution are notably associated with vascular damage. This was exemplified by positive long-term associations between NOx, ^•^NO_2_, PM_2.5_, and PM_10_ exposure and the risk of vascular dementia. In contrast, the relationships with AD and senile dementia were less evident [246]. Consistently with these findings, after accounting for potential confounding factors and other air pollutants, a Taiwanese study established associations between PM_10_, CO, and ^•^NO_2_ exposure and vascular dementia [247]. These relationships were further confirmed by additional studies conducted in Sweden and Canada, which indicated that exposure to multiple air pollutants may elevate the risk of vascular dementia and AD [248,249,250,251].

Notably, as traffic is a primary source of air pollution in urbanized areas and is also associated with increased noise pollution, research endeavors have explored the combined effects of air and noise pollution on the risk of dementia. However, these investigations did neither uncover any substantial evidence linking road traffic noise exposure with dementia risk, nor did they identify any significant interactions between noise and air pollutants that would modulate the risk of dementia [252,253].

In a cohort of 161,808 postmenopausal women aged 50–79 years at baseline, a nested case–control study of 256 ALS deaths and 2486 matched controls was performed [254]. Overall, there was only suggestive evidence for an association between PM_10–2.5_ and ALS deaths. More conclusive evidence was found in a case–control study involving 51 sporadic ALS cases and 51 matched controls [255]. It was demonstrated that residential exposure to aromatic solvents was associated with a higher risk of ALS among cases than controls (OR 5.03, 95% CI 1.29–19.53). Likewise, in a cohort from the Netherlands, the risk of ALS was higher in subjects with increased exposure to PM_2.5_ (OR 1.67, 95% CI 1.27–2.18), ^•^NO_2_ (OR 1.74, 95% CI 1.32–2.30), and NOx (OR 1.38, 95% CI 1.07–1.77) [256]. Further studies support the hypothesis of an association between air pollution and ALS risk [257,258,259,260].

There is not much quality literature on the association between multiple sclerosis and air pollution. A study from Iran found an association of PM_10_ and SO_2_ with higher risk of multiple sclerosis [261]. A study from Lombardy in northern Italy found a significant association between PM_10_ and hospital admissions for multiple sclerosis with an increase of 42% (RR 1.42, 95% CI 1.39–1.45) on the days preceded by one week with PM_10_ levels in the highest quartile [262]. Relapse in multiple sclerosis patients was also found to be associated with air pollution [263].

### 4.2. Transportation Noise and Neurodegenerative Diseases

In a study comprising 5227 participants aged 65 years or older from the Chicago Health and Aging Project, residential noise exposure was linked to a 36% higher likelihood of prevalent mild cognitive impairment (OR 1.36, 95% CI 1.15–1.62) and a 29% higher likelihood of AD (OR 1.29, 95% CI 1.08–1.55) for every 10 dB(A) increase in noise levels [264]. Noise exposure was also associated with lower global cognitive performance, particularly in perceptual speed, although it did not consistently lead to cognitive decline.

Similarly, data from the German Heinz Nixdorf Recall study indicated that exposure to traffic-related noise was linked to decreased global cognitive scores and an increase in mild cognitive impairment [265]. Notably, these associations were more pronounced in former and current smokers, suggesting that lifestyle risk factors may potentiate the adverse cognitive effects of noise exposure [266]. Importantly, the study also revealed that air pollution and road traffic noise exposure may interact synergistically to impact cognitive function negatively [267].

In another study involving 288 elderly women from the German SALIA study, road traffic noise exposure was associated with impaired total cognition and the constructional praxis domain, and these effects remained significant even after adjusting for air pollution exposure [268]. The Irish Longitudinal Study on Aging found that road traffic noise exposure hurts executive function [269]. Additionally, in a study involving 1612 elderly Mexican-American participants from Sacramento, there was suggestive evidence that traffic noise exposure increased the risk of dementia and cognitive impairment [270]. Interestingly, a subsequent study demonstrated that metabolic dysfunction, such as hyperglycemia or low HDL-cholesterol, could modify the influence of traffic-related air pollution and noise exposure on these outcomes [271].

Linares et al. reported a short-term association between traffic noise and dementia-related emergency hospital admissions risk in Spain [272]. In contrast, a study by Andersson et al. did not find any significant effect of road traffic noise exposure, independently or in conjunction with traffic-related air pollution, on the risk of dementia in a cohort of 1721 subjects [252]. In a larger study of 130,978 subjects in London, the relationship between night-time traffic noise exposure and the incidence of dementia became statistically insignificant when multiple air pollutants were considered in the analysis [253].

Likewise, in a large Canadian study involving approximately 678,000 individuals, road proximity and air pollution were positively associated with the risk of dementia and PD, while noise exposure showed no such relationship with these outcomes [273]. Most recently, a nationwide study in Denmark that included nearly 2 million adults aged 60 years or older examined the impact of road traffic and railway noise exposure on the incidence of dementia. The results indicated that both road traffic noise and railway noise exposure were associated with an increased risk of AD. Notably, only road traffic noise exposure, but not railway noise, was linked to a higher risk of vascular dementia [274]. Data from the Danish Nurse Cohort study also suggested an increased risk of death from dementia in response to road traffic noise (HR 1.12, 95% CI 0.90–1.38) [275].

There is a general lack of studies addressing other neurodegenerative diseases, such as multiple system atrophy, frontotemporal dementia, and Huntington’s disease. It is mostly the low prevalence of these diseases that does not provide enough patients for respective exposure–disease association studies or renders them underpowered. A good example of this is a study from UK that found a trend in frontotemporal dementia and air pollution exposure, however, without reaching significance due to the low number of patients [276]. In addition, the major genetic component of Huntington’s disease rather speaks against an association with environmental risk factors.

## 5. Conclusions

Environmental stressors such as noise and air pollution gain more and more attention in the modern urbanized and industrialized world. The fact that the human population is aging rapidly due to better healthcare, nutrition, and sanitation and that neurodegenerative diseases are considered ailments of the elderly, warrants a better understanding of the impact of environmental stressors on neurodegenerative diseases. This review has provided an overview of the most common neurodegenerative diseases and their pathomechanisms. We also highlighted epidemiological evidence for an association of noise and air pollution exposure to neurodegenerative diseases and mechanistic data connecting neurodegenerative pathophysiology to noise and air pollution exposure through neuroinflammation and oxidative stress. Importantly, inflammation and oxidative stress can promote each other in a vicious cycle (Figure 2), further progressing neurodegenerative diseases. More high-quality research is urgently needed, as more mechanistic and epidemiological data could drive better treatment and preventive strategies. Finally, more and better mechanistic insight and clinical evidence would also force better legislative interventions by implementing stricter limits for noise levels and air pollution concentrations. So far, the mitigation of adverse health effects by noise is mostly based on lowering the exposure levels, e.g., by speed limits, noise-isolating windows, lower noise-emitting car engines and tires, train breaks, and optimized aircraft starting/landing protocols. With respect to air pollution, lowering the legal limits is the preferred strategy, e.g., by better filtration systems for combustion emissions, traffic bans in city centers, and the regulation of home heating strategies. In addition, personal measures are popular such as face masks and household air filter systems. A detailed list of mitigation measures against noise and air pollution can be found in two recent review articles [1,2].

## Figures and Tables

**Figure 1 antioxidants-13-00266-f001:**
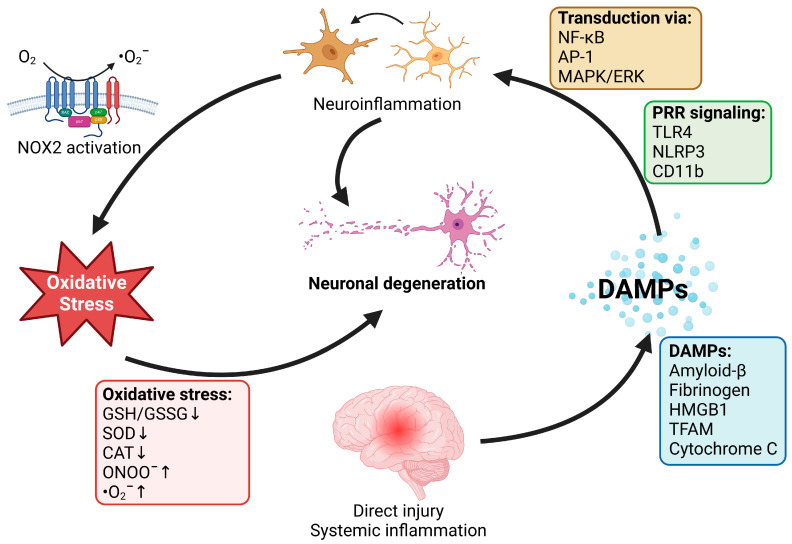
Interplay between neuroinflammation and oxidative stress in neurodegeneration. Direct injury to the neuronal tissue or systemic inflammatory response trigger the activation of damage-associated molecular patterns (DAMPs—such as amyloid-beta, fibrinogen, high mobility group box 1 (HMGB1) protein, mitochondrial transcription factor A (TFAM) and cytochrome c). DAMPs signal activation of microglia (resident macrophages) through pattern recognition receptors (PRRs), such as complement receptor 3 (CD11b), Toll-like receptor 4 (TLR4), and NLR family pyrin domain containing 3 (NLRP3) inflammasome. These signals are transduced in microglia by nuclear factor κB (NF-κB), activator protein 1 (AP-1), and mitogen-activated protein kinases (MAPKs)/extracellular signal-regulated kinase (ERK). Activated microglia produce superoxide (^•^O_2_^−^) through NADPH oxidase (NOX-2), promoting oxidative stress. Oxidative stress can be based on lower antioxidant defense (reduced/oxidised glutathione—GHS/GSSG; superoxide dismutase—SOD; catalase—CAT), and an increase in reactive oxygen and nitrogen species (peroxynitrite—ONOO^−^ and ^•^O_2_^−^) formation. Both neuroinflammation and oxidative stress have a detrimental effect on neuronal tissue, causing neuronal death and promoting neurodegeneration. Adapted from [66] under Creative Common CC BY license. Created with BioRender.com.

**Figure 2 antioxidants-13-00266-f002:**
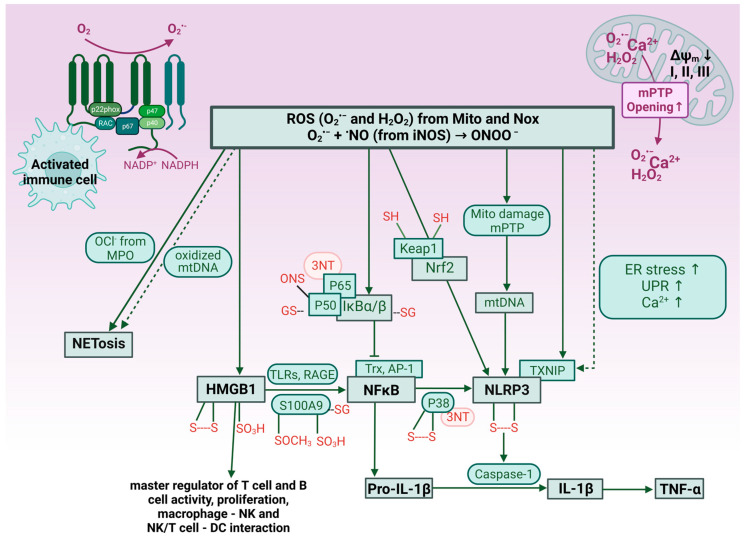
Overview on redox-dependent exacerbation of inflammatory processes and pathways. Primary sources of superoxide are the phagocytic NADPH oxidase in activated immune cells and damaged mitochondria. Superoxide either dismutates to hydrogen peroxide or reacts with nitric oxide from iNOS to form peroxynitrite (ONOO^−^). These ROS can cause redox modifications in master regulators of inflammation such as HMGB1, NF-κB, and NLRP3 or of central inflammatory processes such as NETosis. Also upstream regulators of these processes are redox-regulated. The red-groups indicate redox modifications. S—S, disulfide bridge; SO_3_H, sulfonic acid; SOCH_3_, oxidized methionine; SG, S-glutathionylation; 3NT, 3-nitrotyrosine; SNO, S-nitrosothiol; SH, reduced thiol. Other abbreviations are as follows: ER, endoplasmic reticulum; UPR, unfolded protein response; mPTP, mitochondrial permeability transition pore; mtDNA, mitochondrial DNA; Trx, thioredoxin; AP-1, activator protein 1 transcription factor; IkB, nuclear factor of kappa light polypeptide gene enhancer in B-cells inhibitor; TLR, Toll-like receptor; RAGE, receptor for advanced glycation end products; OCl^−^, hypochlorite; MPO, myeloperoxidase; Keap1, Kelch-like ECH-associated protein 1; Nrf2, nuclear factor erythroid 2-related factor 2; Casp-1, caspase 1; Nox, NADPH oxidase. Scheme was generated from data reported in [65,158]. Created with BioRender.com.

**Figure 3 antioxidants-13-00266-f003:**
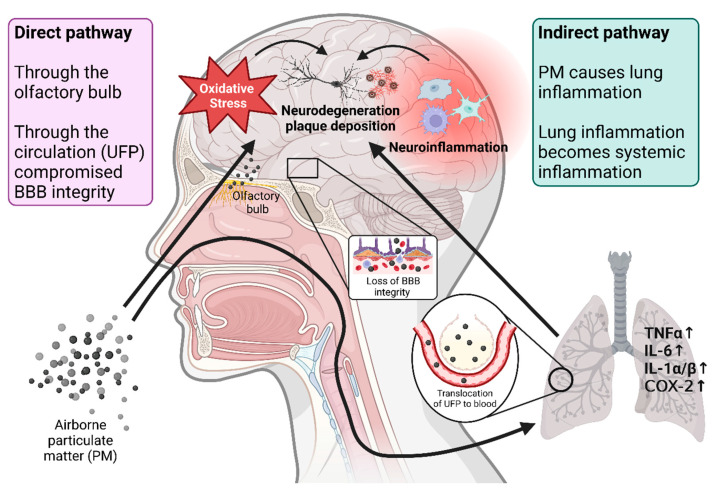
Mechanisms of air pollution-induced neuroinflammation and oxidative stress. Air pollution can cause neuroinflammation and oxidative stress through two pathways: direct and indirect. Air pollution components directly interact with the neuronal tissue in the direct pathway. This can happen by direct contact of air pollution components, such as particulate matter (PM), with the nerves in the olfactory bulb, helping them translocate deeper into the neuronal tissue. Another way is for ultrafine particles (UFP) with a diameter in the nanometer range to enter the systemic circulation after inhalation, causing disturbance of the blood–brain barrier (BBB) and reaching the cerebral tissue. The indirect pathway starts by initiating local inflammation in the lung, the first point of impact for inhaled air pollution components, which then gains systemic character. This systemic inflammation, in the form of circulating cytokines and pro-inflammatory mediators (such as tumor necrosis factor-alpha—TNFα; interleukins 1α/β and 6—IL-1α/β/6; and cyclooxygenase 2—COX-2), spreads to the brain where it promotes activation of microglia (resident macrophages). Both of these pathways result in neuroinflammation and cerebral oxidative stress, which promotes the development and progression of neurodegenerative diseases. Created with BioRender.com.

**Figure 4 antioxidants-13-00266-f004:**
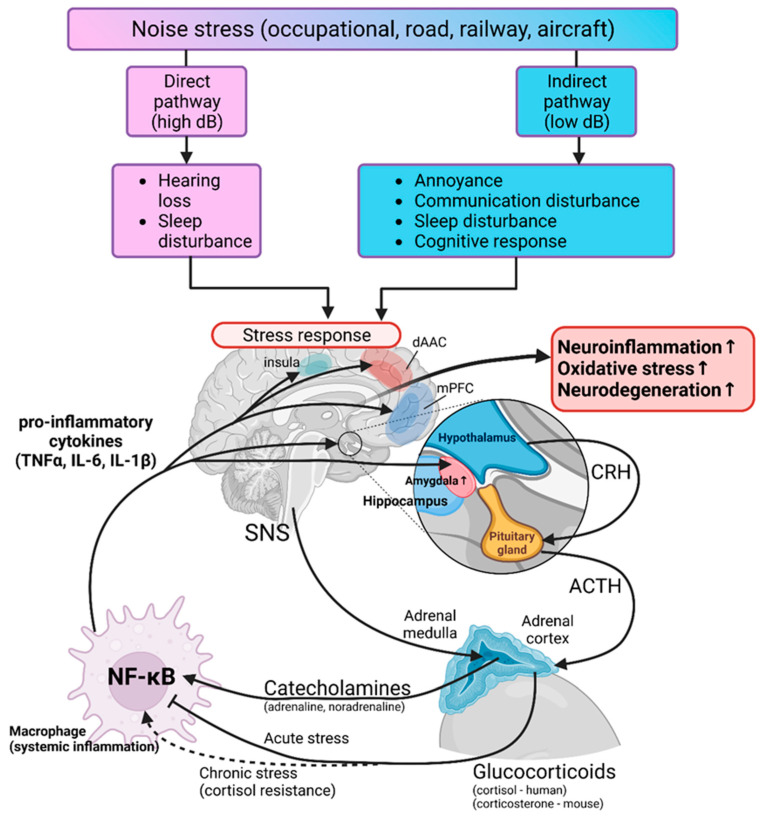
Mechanisms of noise-induced neuroinflammation and oxidative stress. Noise can cause neuroinflammation and oxidative stress through two pathways: direct and indirect. In the direct pathway, noise causes inner ear damage, hearing loss, and sleep deprivation, while in the indirect pathway, it creates a state of annoyance and disturbs sleep and communication. Both of these pathways activate the stress response through the activation of the hypothalamic–pituitary–adrenal (HPA) axis and the activation of the sympathetic nervous system (SNS). The activation of the HPA axis results in the release of CRH (corticotropin-releasing hormone) and ACTH (adrenocorticotropic hormone), further promoting the release of glucocorticoids from the adrenal cortex. The activation of the SNS results in secretion of catecholamines from the adrenal medulla. Both the HPA axis and the SNS activation result in systemic inflammation signaling through macrophage activation and cytokine release (such as tumor necrosis factor-alpha—TNFα; and interleukin 6/1β—IL-6/1β), further activating neuroinflammation and oxidative stress. Abbreviations: NF-κB (nuclear factor kappa-light-chain-enhancer of activated B cells), dAAC (dorsal anterior cingulate cortex), mPFC (medial prefrontal cortex). Adapted from [218]. Created with BioRender.com.

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
