# Peer review of "Crosstalk between Oxidative Stress and Inflammation Caused by Noise and Air Pollution—Implications for Neurodegenerative Diseases"

_antioxidants, 2024, doi:10.3390/antiox13030266_

Round 1

Reviewer 1 Report

Comments and Suggestions for Authors

The current review summarize works on noise and air pollution induced oxidative stress and inflammation and human neurodegenerative diseases. This is an interesting topic to be discussed. 

However, some concerns are raised:

1, The authors only discussed on AD, PD and ALS. However, human neurodegenerative diseases include many disorders, including MSA, FTD, Huntington's disease, prion disease and spinocerebellar ataxias. How noise and air pollution can be pathogenic factors for these disorders?

2, Most of ROS linked to human neurodegenerative diseases can be generated from mitochondria. In PD, the dopamine related ROS generation plays a pathogenic role. So the authors discussion on ROS generated from inflammation have overlooked the pathological ROS production beyond inflammation. 

3, for the roles of microglia and astrocytes related brain inflammation related to neurodegenerative disease, the authors should discuss more, including some novel findings.

4, so far whether the noise and air pollution can cause neurodegenerative diseases are still under debate. Most evidence are only correlative postulations from animal works, therefore the authors should make the statement and discuss this issue.

 minor issues:

1, the authors use abbreviation for AD and PD in the introduction and then use full name in the later sections. So it seems to be confusing without careful proof reading.  

Author Response

The current review summarize works on noise and air pollution induced oxidative stress and inflammation and human neurodegenerative diseases. This is an interesting topic to be discussed. 

Answer: We thank the reviewer for their favourable general evaluation of our manuscript.

However, some concerns are raised:

1, The authors only discussed on AD, PD and ALS. However, human neurodegenerative diseases include many disorders, including MSA, FTD, Huntington's disease, prion disease and spinocerebellar ataxias. How noise and air pollution can be pathogenic factors for these disorders?

Answer: We thank the reviewer for his/her comment. AD, PD and ALS were chosen to be the primary subjects of this manuscript because of their high prevalence and their observed correlation with noise and air pollution. Searching the PubMed and Google Scholar databases, no published work was found that associated noise to either MSA or FTD. For air pollution only one study looked at the correlation with FTD (DOI: 10.1016/j.envres.2022.112895) and found no statistical significance. No studies looked at the association between air pollution and MSA. Huntington’s disease also lacks any epidemiological data on correlation with noise and air pollution, with only one animal study providing very weak noise association (DOI: 10.3390/brainsci13040573). Also, Huntington’s diseases is mainly heritable, providing no incentive to associate it to the environmental risk factors.

We have now added a statement on page 15 explaining the lack of other neurodegenerative diseases in the manuscript: “There is a general lack of studies addressing other neurodegenerative diseases, such as multiple system atrophy, frontotemporal dementia and Huntington’s disease. It is mostly the low prevalence of these diseases that does not provide enough patients for respective exposure-disease-association studies or renders them underpowered. A good ex-ample of this is a study from UK that found a trend in frontotemporal dementia and air pollution exposure, however, without reaching significance due to the low number of patients. In addition, the major genetic component of Huntington’s disease rather speaks against an association with environmental risk factors.”

We have also added a paragraph on multiple sclerosis: “There is not much quality literature on the association between multiple sclerosis and air pollution. A study from Iran found an association of PM10 and SO2 with higher risk of multiple sclerosis. A study from Lombardy in northern Italy found a significant association between PM10 and hospital admissions for multiple sclerosis with an increase of 42% (RR 1.42, 95% CI 1.39–1.45) on the days preceded by one week with PM10 levels in the highest quartile. Relapse in multiple sclerosis patients was also found to be associated with air pollution.”

2, Most of ROS linked to human neurodegenerative diseases can be generated from mitochondria. In PD, the dopamine related ROS generation plays a pathogenic role. So the authors discussion on ROS generated from inflammation have overlooked the pathological ROS production beyond inflammation. 

Answer: We thank the reviewer for his/her insightful comment. We have already mentioned the role of mitochondria in PD, see page 5: “The involvement of mitochondria was also observed in PD patients with a reduction in respiratory chain complexes activity, and inhibition of complex I, e.g., by rotenone, was previously shown to cause Parkinsonism, which was even used as an animal model of PD. Therapies with coenzyme Q10, an important molecule for the function of the mitochondrial respiratory chain, showed limited but positive outcomes on the progression of PD, although later meta-analysis showed no improvement compared to placebo.”

The role of dopamine in ROS generation was also discussed on page 5: “Interestingly, some research shows that protein carbonyls might come from L-DOPA, a therapeutic drug used to treat dopamine deficiency in PD, as it can have pro-oxidant properties via hydroxyl radicals. L-DOPA could also provide the substrates for mitochondrial monoamine oxidases, e.g., dopamine and other catecholamines, with subsequent hydrogen peroxide formation, which was therapeutically exploited using monoamine oxidase inhibitors in PD patients.” We just added the word “mitochondrial” to make this clearer.

On page 9 we extended the sentence “Exposure to PM was also associated with known pathomechanisms of neurodegenerative diseases, also including mitochondrial dysfunction and reactive oxygen species formation.” to make clear that PM via mitochondrial ROS formation could contribute to neurodegenerative diseases.

In figure 2 we have highlighted mitochondrial ROS formation in the cross-activation of inflammatory pathways.

Although we have mentioned non-inflammation sources of ROS throughout the manuscript, we agree with the reviewer on the need for a clearer explanation. We have now added a sentence on page 3/4 to clarify this: “It is important to keep in mind that neuroinflammation is not the only source of cerebral oxidative stress, and that other prominent sources like the mitochondria and the dopamine metabolism can significantly contribute to neurodegeneration.”

3, for the roles of microglia and astrocytes related brain inflammation related to neurodegenerative disease, the authors should discuss more, including some novel findings.

Answer: We thank the reviewer for their comment. Although we already mentioned microglia here and there, we have now added another paragraph that includes some novel findings from the field of neurodegeneration associated microglia:

“Since microglia play a central role in neuroinflammation, it is important to consider them in the light of novel findings associating their function with neurodegenerative diseases. Advances in single cell RNA sequencing revealed that the traditional M1/M2 polarization paradigm is not detailed enough to explain the intricate differences in specific microglia gene expressions associated with neurodegenerative diseases. Microglia responding to Aβ have a distinct expression profile envisaged by upregulation of APOE, TREM2, GPNMB, and are CD163-positiv. Similar specific gene expression was found for phosphorylated tau protein, like GRID2, and for α-synuclein, like IL1B, GPNMB and HSP90AA1, in PD patients.”

The important role of astrocytes in neuroinflammation has also been addressed in a new paragraph:

“Astrocytes are part of the glymphatic system that is responsible for clearing protein plaques, such as Aβ, but they are also involved in neurodegenerative disease induced neuroinflammation. Similarly to microglia, astrocytes express neurodegenerative disease associated genes that tend to promote inflammatory processes, like CD44, GFAP and HSPB1, observed in PD, Huntington’s disease, and AD.”

4, so far whether the noise and air pollution can cause neurodegenerative diseases are still under debate. Most evidence are only correlative postulations from animal works, therefore the authors should make the statement and discuss this issue.

Answer: We agree with the reviewer that there is no clear relationship between the onset of neurodegenerative diseases and noise/PM exposure – although the epidemiological data presented by us support this hypothesis. We have highlighted this important point with a sentence on page 12: “Animal data on the relationship between the onset of neurodegenerative diseases and noise/PM exposure are mostly correlative in nature and large clinical cohorts or epidemiological studies are scarce. Therefore, the present topic is still under debate and high quality human research is urgently needed.”

 minor issues:

1, the authors use abbreviation for AD and PD in the introduction and then use full name in the later sections. So it seems to be confusing without careful proof reading.  

Answer: We have now corrected this.

Reviewer 2 Report

This manuscript summarizes the effects of air pollutants and noise on the levels of oxidative stress resulting in the onset and/or progression of neurodegenerative diseases in detail. The authors have reviews numerous previously published papers; however, more recent studies should be covered in this review article.

1.      Although this manuscript cites 255 references, only 82 references (ca. 32%) have been published in the last 5 years (from 2019 to 2024). the authors should focus on the latest findings and provide a comprehensive review of the most recent researches with a suggestion of new perspective for future research.

2.      This manuscript summarizes the effects of oxidative stress (induced by PM and noise) in pathogenesis of neurodegenerative diseases; however, a more in-depth exploration of the specific contributions of individual ROS (or RNS) to elevating oxidative stress levels would enhance the overall clarity and depth of the discussion.

3.      Please provide the strategy (individual and/or governmental approaches) to regulate oxidative stress from air pollutants and noise in conclusion section.

Author Response

This manuscript summarizes the effects of air pollutants and noise on the levels of oxidative stress resulting in the onset and/or progression of neurodegenerative diseases in detail. The authors have reviews numerous previously published papers; however, more recent studies should be covered in this review article.

Detail comments

  1. Although this manuscript cites 255 references, only 82 references (ca. 32%) have been published in the last 5 years (from 2019 to 2024). the authors should focus on the latest findings and provide a comprehensive review of the most recent researches with a suggestion of new perspective for future research.

Answer: Although we appreciate the sentiment of the reviewer, the percentage of recent references is not a general marker for the relevance of a manuscript. In addition, looking at a slightly different time window (2016-2024) reveals that 53% of our cited references were published within the last 8 years – which is far from being outdated.

For the first part of the review, we have explored long existing concepts like oxidative stress and inflammation, and provided a lot of background for neurodegenerative disease which have been known for decades. Accordingly, it is just fair to pay credit to the landmark publications in these fields.

For the second part of the review, the impact of noise and air pollution exposure on the onset and progression of neurodegenerative diseases was our main focus. This is a highly specialized research field, where anyhow not that many publications exist – so in many cases there was not a choice between old or new references but we took what is available.

  1. This manuscript summarizes the effects of oxidative stress (induced by PM and noise) in pathogenesis of neurodegenerative diseases; however, a more in-depth exploration of the specific contributions of individual ROS (or RNS) to elevating oxidative stress levels would enhance the overall clarity and depth of the discussion.

Answer: We thank the reviewer for their question. The impact of individual ROS, more specifically H2O2, ·O2- and ·NO, and their impact on the most important pro-inflammatory pathways is presented in detail in figure 2. It is generally not possible to address the individual contribution of each ROS species, unless they were individually measured by specific dyes or techniques – which is mostly not the case for environmental exposure studies. In order to make clear which RONS species we talk about, we have added the following paragraph on page 4: “The RONS that we talk about in the subsequent sections, are mostly those produced from NADPH oxidases during inflammation and by mitochondria upon stress challenges (superoxide and hydrogen peroxide) or direct physicochemical effects from air pollutants (e.g. nano-sized combustion-derived particles). The term RONS also comprises the major product of superoxide and nitric oxide, peroxynitrite. Besides these most important RONS, there may be some minor contribution by hydroxyl radicals that are directly formed from transition metal reactions with peroxides or other redox-active chemicals on the surface of air pollution particles. More details on the species formed and their sources can be found in recent overviews.”

  1. Please provide the strategy (individual and/or governmental approaches) to regulate oxidative stress from air pollutants and noise in conclusion section.

Answer: Unfortunately, it is not possible to regulate oxidative stress through government intervention, as it is a biochemical state, rather than a disease or behavioural risk factor. What is done by governmental agencies is to reduce and enforce the exposure limits of the population. We have now briefly mentioned some of the most common strategies in the Conclusions and have cited respective references:

“ So far mitigation of adverse health effects by noise is mostly based on lowering the expo-sure levels, e.g. by speed limits, noise isolating windows, lower noise emitting car engines and tyres, train breaks, optimized aircraft starting/landing protocols. With respect to air pollution, lowering the legal limits is the preferred strategy, e.g. by better filter systems for combustion emissions, traffic bans in city centers, and regulation of home heating strategies. In addition personal measures are popular such as face masks and household air filter systems. A detailed list of mitigation measures against noise and air pollution can be found in two recent review articles.”

Round 2

Reviewer 1 Report

The revised manuscript has been improved significantly. However, there is one point needs to be addressed. Th author has discussed the ROS generation from L-DOPA. Recent achievements highly the pathogenic roles of endogenous dopamine in PD. So the authors should add discussion on dopamine related ROS generation, especially dopamine -o-quinone induced toxicity, rather than only on L-DOPA. The recent paper (Role of dopamine in the pathophysiology of Parkinson's disease. Translational Neurodegeneration. 2023) should be cited and discussed. 

NA

Author Response

Comment: The revised manuscript has been improved significantly. However, there is one point needs to be addressed. Th author has discussed the ROS generation from L-DOPA. Recent achievements highly the pathogenic roles of endogenous dopamine in PD. So the authors should add discussion on dopamine related ROS generation, especially dopamine -o-quinone induced toxicity, rather than only on L-DOPA. The recent paper (Role of dopamine in the pathophysiology of Parkinson's disease. Translational Neurodegeneration. 2023) should be cited and discussed. 

Answer: We highly appreciate the reviewer comment and have addressed this point in full detail by adding the additional text on page 5 of the manuscript:

"Another prominent feature of dopamine is its ability to undergo auto-oxidation, or metal-catalyzed oxidation reactions, resulting in formation of dopamine quinones [103]. The short lived dopamine-o-quinone is created together with O2-, and can undergo further oxidation to form aminochrome [104]. Both dopamine-o-quinone and aminochrome are highly reactive and form protein adducts on the cysteine, lysine and tyrosine residues, causing specific damage to mitochondria and thereby worsening the PD pathology. "